# *TP53* Targeted Deep Sequencing of Cell-Free DNA in Esophageal Squamous Cell Carcinoma Using Low-Quality Serum: Concordance with Tumor Mutation

**DOI:** 10.3390/ijms22115627

**Published:** 2021-05-26

**Authors:** Dariush Nasrollahzadeh, Gholamreza Roshandel, Tiffany Myriam Delhomme, Patrice Hodonou Avogbe, Matthieu Foll, Farrokh Saidi, Hossein Poustchi, Masoud Sotoudeh, Reza Malekzadeh, Paul Brennan, James Mckay, Pierre Hainaut, Behnoush Abedi-Ardekani

**Affiliations:** 1Digestive Oncology Research Center, Digestive Disease Research Institute, Shariati Hospital, Tehran University of Medical Sciences, Tehran 14117-13135, Iran; neshelid@fellows.iarc.fr (D.N.); saidif@ams.ac.ir (F.S.); h.poustch@gmail.com (H.P.); masoud.sotoudeh.md@gmail.com (M.S.); dr.reza.malekzadeh@gmail.com (R.M.); 2Genomic Epidemiology Branch, International Agency for Research on Cancer/World Health Organization (IARC/WHO), 69000 Lyon, France; tiffany.delhomme@irbbarcelona.org (T.M.D.); patrice.avogbe@gmail.com (P.H.A.); follM@iarc.fr (M.F.); brennanp@iarc.fr (P.B.); mckayj@iarc.fr (J.M.); 3Golestan Research Center of Gastroenterology and Hepatology, Golestan University of Medical Sciences, Gorgan 49177-44563, Iran; drroshandel@goums.ac.ir; 4Institute for Research in Biomedicine (IRB Barcelona), Barcelona Institute of Science and Technology, 08036 Barcelona, Spain; 5Institute for Advanced Biosciences, Inserm 1209 CNRS 5309 UGA, 38700 Grenoble, France; pierre.hainaut@univ-grenoble-alpes.fr

**Keywords:** circulating cell-free DNA, liquid biopsy, circulating tumor DNA, esophageal squamous cell carcinoma, *TP53*, tumor mutation, variant caller, deep sequencing, neoantigen

## Abstract

Circulating cell-free DNA (cfDNA) is emerging as a potential tumor biomarker. CfDNA-based biomarkers may be applicable in tumors without an available non-invasive screening method among at-risk populations. Esophageal squamous cell carcinoma (ESCC) and residents of the Asian cancer belt are examples of those malignancies and populations. Previous epidemiological studies using cfDNA have pointed to the need for high volumes of good quality plasma (i.e., >1 mL plasma with 0 or 1 cycles of freeze-thaw) rather than archival serum, which is often the main available source of cfDNA in retrospective studies. Here, we have investigated the concordance of *TP53* mutations in tumor tissue and cfDNA extracted from archival serum left-over from 42 cases and 39 matched controls (age, gender, residence) in a high-risk area of Northern Iran (Golestan). Deep sequencing of *TP53* coding regions was complemented with a specialized variant caller (Needlestack). Overall, 23% to 31% of mutations were concordantly detected in tumor and serum cfDNA (based on two false discovery rate thresholds). Concordance was positively correlated with high cfDNA concentration, smoking history (*p*-value = 0.02) and mutations with a high potential of neoantigen formation (OR; 95%CI = 1.9 (1.11–3.29)), suggesting that tumor DNA release in the bloodstream might reflect the effects of immune and inflammatory context on tumor cell turnover. We identified *TP53* mutations in five controls, one of whom was subsequently diagnosed with ESCC. Overall, the results showed that cfDNA mutations can be reliably identified by deep sequencing of archival serum, with a rate of success comparable to plasma. Nonetheless, 70% non-identifiable mutations among cancer patients and 12% mutation detection in controls are the main challenges in applying cfDNA to detect tumor-related variants when blindly targeting whole coding regions of the *TP53* gene in ESCC.

## 1. Introduction

Circulating tumor DNA is but a tiny fraction of cfDNA in blood circulation. CfDNA can be extracted from plasma or serum, although plasma is widely recommended as the preferable standard media for studying cfDNA [1,2]. However, widespread use of serum for medical workups in clinical and hospital biobanks have made it more easily available than plasma in retrospective case-control studies, and it is the sole available resource in many population-based studies. It is therefore important to investigate whether cfDNA may be successfully retrieved and sequenced from archival collections.

Esophageal squamous cell carcinoma (ESCC), with nearly 440,000 deaths annually [3], has a dismal prognosis [4]. Symptomatic diagnosis is usually late.Due to space availability in the mediastinum, lack of serosa, and flexible tubal structure, tumor mass in the esophagus can expand silently until obstructive symptoms appear. As a result, 5-year survival rates are low (5–15%) but significantly improve when diagnosis is made at early stages [5]. Notably, a 95% 5-year survival rate has been reported given early diagnosis [6]. Despite several ongoing efforts for early detection of ESCC in endemic areas [7], a minimally invasive and feasible approach has yet to be developed. Liquid biopsy and its cfDNA component have shown encouraging features in studying treatment response among ESCC patients [8].

*TP53* is the most frequently mutated gene in ESCC, with about 90% of ESCC tumors carrying one or more *TP53* mutations in the Asian esophageal cancer belt [9,10]. *TP53* mutations are detectable in the dysplastic squamous mucosa, highlighting its potential as an early biomarker for esophageal squamous dysplasia and ESCC [11]. Studies have evaluated and reviewed the diagnostic and prognostic value of mutation detection in liquid biopsies of ESCC using next-generation sequencing [12,13,14]. Though the concordance fraction of serum cfDNA for tumor mutations has not been investigated as thoroughly [15], to our knowledge, published studies have not reported the diagnostic value of serum cfDNA in ESCC.

Over the past few decades, several large case-control studies on ESCC epidemiology in endemic areas have collected serum samples. These archived biosamples may not meet pre-analytical standards for cfDNA evaluation. However, they represent a unique resource associated with large-scale, well-documented studies, which are the cornerstone of our current knowledge on ESCC epidemiology in endemic areas. Therefore, it is of particular interest to investigate their diagnostic value in cfDNA studies.

Here we have applied exon-targeted deep sequencing, coupled with a stringent bioinformatics approach, to detect *TP53* mutations in cfDNA from left-over archival sera collected in a case-control study conducted between 2002–2008 in Golestan (Northern Iran). Serum cfDNA *TP53* variants were analyzed in 42 ESCC and compared to matched tumor DNA variants previously identified by Sanger sequencing, providing a concordance ratio between cfDNA and tumor DNA. We also assessed cfDNA *TP53* variants in 39 matched cancer-free patients.

## 2. Results

A total of 42 ESCC cases and 39 matched controls were initially included in this study. In 22 cases, tumor location was in the mid-third of the esophagus. The bulk of tumor was in the lower third and upper third of the esophagus in 18 and 2 cases, respectively. The mean age of cases and controls were 62 and 64 years old. Half of the controls and 52% of cases were females. The proportion of smokers among cases (23.8%) and controls (20.5%) was comparable. The mean of left-over serum volume for cfDNA extraction was 581 µL among cases and 880 µL among controls (*p*-value = 0.00001). After excluding 3 outliers for cfDNA level (>100 ng/mL) the mean concentration of cfDNA among neighborhood controls and cases were 1.62 and 1.18 ng/mL (*p*-value = 0.06), respectively. Targeted deep sequencing readouts were analyzed using Needlestack, a multi-sample variant caller designed for the detection of low abundance mutations. Mean sequencing coverages for first and second technical duplicates were 557,735 and 596,654 reads among cases, respectively, and 572,085 and 532,498 reads among controls, respectively. Total reads in duplicates were less than 100 for one ESCC case and three controls. Figure 1 depicts the distribution of cfDNA fragments in an ESCC case after applying size-selection using magnetic beads.

We observed a concordance between *TP53* variants previously detected in FFPE tumor biopsy and those detected by deep-sequencing in serum cfDNA in 10 cases (24% concordance), among which, in five cases, *TP53* variants were observed in both technical duplicates. By decreasing the Q-value threshold from 50 to 20, cfDNA variants in additional five ESCC cases were in concordance with tumor mutations (36%). Among them, in eight cases, *TP53* variants were observed in both technical duplicates. Figure 2 depicts the effect of varying Q-value thresholds on detecting total *TP53* variants in cfDNA among cases and controls, regardless of *TP53* alteration in tumor tissue.

We searched for *TP53* variants in cfDNA, matching the pool of positions and base changes detected in *TP53* across all ESCC tumors included in our study. Among the 42 cases, cfDNA variants matched with tumor variants in 13 patients (31% concordant, 8 duplicate). Among the 39 controls, we observed five single *TP53* variants in six age-sex-matched neighborhood controls. None of these *TP53* variants were detected in both technical duplicates. During follow-up, one of six controls with 7577079C>T variant was diagnosed with ESCC less than a year after enrollment. Another control, harboring the same mutation in cfDNA, was alive without any sign of malignancy 15 years after enrollment. The cause of mortality amongst remaining controls with *TP53* variants in cfDNA (7577121G>T, 7577509C>T, 7577536T>A, 7577580T>C) was not cancer-related and the death date was between 6 and 10 years after enrollment.

Table 1 compares primary exposure data, total DNA, and sequencing performance among cases and controls. Drinking and smoking are two major risk factors for ESCC. In the study area, drinking habits are uncommon, and opium use is an established risk factor for ESCC. A total of 63% of ESCC cases with detected concordant cfDNA and 29% of cases without detectable concordant cfDNA were either tobacco or opium users (*p*-value = 0.04).

Figure 3 depicts the positions of *TP53* variants in cfDNA among healthy controls and ESCC cases. The genomic positions of *TP53* mutations in our series of ESCC tumors were mostly in exons 5–8. *TP53* variants in serum cfDNA showed a similar pattern. Among five cfDNA *TP53* variants identified in controls, 7577509C>T (E258K) was considered pathogenic (CLINSIG database) with a REVEL score of 0.96. The mean allelic fraction of variants in controls’ cfDNA was 0.19% and ranged from 0.01% to 0.42%. After excluding one ESCC case with a higher than 10% allelic fraction, cfDNA allelic fraction among cases with concordant *TP53* mutations ranged from 0.04% to 0.94%, with a mean of 0.27%. The allelic fraction of concordant *TP53* mutations in serum cfDNA was not different between cases and controls (*p*-value = 0.52).

We examined the nucleotide distance between genetic positions of mutations across *TP53* gene in ESCC tumor tissues and categorized them into three groups; less than 5 nucleotides distance between adjacent variants (15 variants), between 5 and 10 nucleotides (7 variants), and more than ten nucleotides distance between variants (22 variants). The mean detection fraction of cfDNA concordant variants decreased from 50% to 20% as genetic coordinates between mutations decreased (Figure 4). The mean of cfDNA allelic fractions was similar among groups (mean allelic fraction = 0.2).

We evaluated the frequency of tumor-specific neoantigen formation using predictions from the TCGA database [17]. In the logistic regression model with the detectability of tumor *TP53* mutations in cfDNA as an outcome, we observed higher detection rate of mutations with higher frequency of neoantigen formation (Appendix A). The associated probability of *TP53* mutation detection in cfDNA did not change when adjusting for the number of HLA subtypes (Table 2, Appendix A).

Table 3 summarizes the characteristics of detected and undetected mutations in cfDNA. We did not observe a statistical difference between variant characteristics in relation to their detectability of tumor mutation in cfDNA.

We completed a follow-up study for 26 controls (out of 39) and 36 ESCC cases (out of 42). Staging data was not available for most cases (66%). For those with available clinical staging data, stage III was the most common. Variant detection in cfDNA did not associate with different survival rates (hazard ratio for detection of concordant cfDNA variants (95%CI): 1.17 (0.49–2.80)). In 15 years of follow-up, four healthy controls developed cancers (two ESCC, one lymphoma and one skin cancer). One control with pathogenic *TP53* variant in cfDNA was diagnosed with ESCC six months after recruitment. Self-report of chronic diseases and inflammatory conditions (arthritis, cardiovascular disease, history of stroke, diabetes, COPD, renal failure, and hepatitis) were absent among controls with detectable *TP53* mutations in the serum cfDNA. In contrast, six controls (19%) with no *TP53* cfDNA variants reported one or more of the above-mentioned conditions.

## 3. Discussion

Using targeted deep sequencing of *TP53* coding regions (exons 2 to 11 and flanking splicing sites), we demonstrated a 24% to 36% concordance fraction between variants detected in cfDNA from archived serum and paired FFPE ESCC tumor tissue. In this study, variants were cataloged as a probable true positive if they met either one of the following criteria: concordant detection in technical duplicates or/and concordance with the variant previously detected by Sanger sequencing in FFPE tumor tissue of the same patient. Overall, the concordance rate reported here using archived serum was comparable to the one reported by others using plasma [19], suggesting that serum may represent a valuable source of cfDNA for detecting tumor-associated gene variants, provided that sufficiently sensitive and specific detection methods are used.

Standard preanalytical conditions for detecting mutations in cfDNA recommend the usage of plasma rather than serum, with a volume of or exceeding 1 mL, with a maximum single cycle of freeze-thaw, and storage at −80 °C [1]. Our study samples were sera with less than 0.8 mL available volume (mean 0.5 mL), kept between 10–13 years at −80 °C in Golestan Biobank and for two years at −20 °C in other research centers, with more than or equal to three cycles of freeze-thaw. Due to the matched study design, preanalytical conditions for cases and controls were similar. The only significant difference was the longer time interval between withdrawing blood and performing centrifuge for neighborhood controls compared to cases in research clinics. The longer time-interval was due to shipment of blood in coolers from distant villages to research center. On-road shipment and longer times prior to serum extraction can cause more lysis and, as a result, we observed marginally but statistically non-significantly higher levels of cfDNA among neighborhood controls compared to ESCC cases. A possible drawback of using serum is the presence of DNA fragments from lysed white blood cells, as a result, serum contains 2–24 times more cfDNA than plasma [20,21,22], resulting in a smaller proportion of tumor-derived DNA fragments in serum cfDNA compared to plasma. To minimize this effect, we included a size selection step before sequencing to reduce the number of long DNA fragments, mostly associated with lysed cells.

Two deep-sequencing approaches have been commonly used to screen for *TP53* mutations in cfDNA. Most studies have focused on a limited number of a priori known point mutations, which allow sequencing of a selected library of certain genomic concordances. Other studies (such as ours) have examined entire coding regions, which allow screening for all possible mutations in the range of DNA targeted. The latter method has the advantage of not requiring prior knowledge of the mutational profile. However, its disadvantage over the former is the possibility of differential coverage of the gene regions, thus potentially missing variants located in the region with lower sequencing coverage. A study in head and neck SCC reported a different concordance fraction of plasma cfDNA when using targeted *TP53* mutation (32%) or *TP53* coding regions (2.7%) [19]. In our study, based on a Q-value threshold of 30 a concordance fraction of 24% was found which is comparable to concordance reported for ESCC in a multigene multi-cancer study [23]. By lowering the Q-value from 50 to 20, the concordance fraction improved to 36%.

Given that *TP53* mutations in our selected cases were located over 4 exons in tumor tissue, our approach of sequencing all coding exons may have decreased the probability of detecting mutations due to unnecessary reduction of sequencing coverage. We observed that more than half of the readings were from the exons without corresponding mutations in the tumor. If we ignored our prior knowledge of mutations in tumors and only approved duplicate variants in cfDNA, the concordance between tumors and cfDNA would be 12% to 19% (based on the Q-value threshold selected). None of the *TP53* variants among controls were duplicates. Based on a handful of studies and a modest number of cases, concordant mutation detection in cfDNA of esophageal cancer cases varies [24], e.g., 14% using 12 gene panels [17].

Serum cfDNA in 74% of our ESCC cases did not show detectable *TP53* variants, suggesting that detection may be associated with specific factors that are unevenly distributed among cases. Independent from the amount of extracted cfDNA, smoking showed a significant positive association with *TP53* detection in cfDNA (*p* = 0.02). Of note, in our study population, tumor *TP53* mutations did not show an association with smoking or chewing tobacco [9]. Thus, the higher rate of concordant *TP53* variants in cfDNA of tobacco smoker/chewer cases suggests that tobacco consumption may be correlated with a higher rate of shedding tumor DNA in the serum. Systemic inflammation induced by smoking may cause increased tumor cell damage and turnover, resulting in the release of tumor DNA in the serum [25]. Alternatively, studies have reported a higher rate of overall clonal hematopoiesis (CH) in smoking-related diseases. In CH, a fraction of white blood cells may carry somatic mutations [26]. Thus, CH and DNA release from lysed white blood cells might account for the origin of some of the observed variants in cfDNA—which are coincidentally the same as those found in the tumor. However, it remains to be determined if CH is due to smoking or the general inflammatory process [27].

A total of 45% of tumor *TP53* mutations in our series were recorded in the TCGA-based neoantigen database [17]. Despite the small sample size, we observed a greater probability of detecting tumor *TP53* mutations in cfDNA among variants with the highest quartile of neoantigen formation. This observation suggests that tumor cells expressing potentially neoantigenic *TP53* variants may be more prone to release tumor DNA in the bloodstream than tumor cells expressing non-antigenic variants. The presence of a neoantigenic variant may specify a different immune microenvironment in the tumor with, perhaps, increased tumor cell turnover and, consequently, tumor DNA release in the bloodstream. As for the association with tobacco usage discussed above, this hypothesis needs further investigation with a larger sample size.

We observed that adjacent variants (less than 5 nt up or downstream) in different samples had a lower chance of being detected in serum cfDNA regardless of their allelic fraction, which explains 26% of missed *TP53* mutations. It could be a random error and we did not determine a reason for it. Among the filtering steps, we removed variants with an allelic fraction of 10 times higher than candidate variants in 5-nt to 10-nt distance from the target candidate [28]. It could indicate that using whole coding region for genes and cancers with prevalent adjacent mutations (<10 nt), may cause some limitations, particularly among variant with too low allelic fraction.

Some studies reported detecting up to 11% of *TP53* variants in cfDNA of non-cancerous controls [29]. In the current study, we did not detect *TP53* variants in both technical duplicates among controls, either through matching for known tumor mutations or blind screening of *TP53* coding regions. We detected non-duplicate *TP53* variants in 6 controls, of which one developed ESCC six months after enrollment. We were unable to verify if the same cfDNA mutation existed in the subsequent esophageal tumor. Given the short time lapse between serum sampling and diagnosis, it is possible that this subject already carried an asymptomatic tumor at the time of recruitment. Alternatively, *TP53* mutations can be detected in a small subset of normal esophageal epithelia, as the result of ongoing exposure to environmental risk factors [30]. Entertaining the possibility of field cancerization [31] due to shared exposure to carcinogens might be a plausible explanation, given that neighborhood controls in this study shared a comparable environment with cases [32].

Likewise, we did not observe *TP53* mutations in cfDNA of controls who subsequently developed other cancers than ESCC. Of note, one of our study’s limitations is that we did not sequence the WBC of these controls to assess the possibility of CH as the origin of the mutation.

Due to the scarcity of DNA of tumor origin in the bloodstream [33], it is a valid argument to address whether the observed concordant mutations were true variants or false-positive findings resulting from our targeted search for mutations [34]. All patients in our series had a tumor diameter of >1 cm. It is estimated that, with such tumor size, tumor DNA would represent between 0.1% and 0.01% of cfDNA. Notably, all detected *TP53* mutations in our studywere identical to those previously found in tumors, with no new *TP53* variant. This observation supports that the variants identified in cfDNA in this study are likelyoriginating from the tumor.

This study had several limitations: a small number of ESCC cases, lack of WBC sequencing, use of different sequencing approaches for the tissue and liquid biopsies, and no validation of NGS-detected cfDNA variants with digital droplet PCR or other methods. Strengths of the study included: age, sex, residence-matched controls in a population-based case-control design, follow-up data of controls on cancer occurrence, application of a sensitive method for variant detection, and use of several filtering steps to decrease the probability of false-positive detection.

In conclusion, based on concordance with tumor mutations, archival serum samples appear useful for detecting targeted ESCC tumor *TP53* mutations in cfDNA. The implications of our findings are important, echoing the message for mutation-based cancer biomarkers when whole coding regions are blindly screened. This limitation is beyond selection of the type of biological samples (serum or plasma). At the same time, we have shown the importance of ultra-sensitive rare variant callers in avoiding recruiting false positive results. We have also portrayed the biological challenges, e.g., adjacent variants, neoantigens, and certain carcinogenic profiles influencing the specificity of mutation detection in cfDNA. This study, as the first attempt at screening whole coding regions of *TP53* amongst one of the highest incident areas for ESCC, reached similar results to those studies which applied the same method in different organs and non-endemic populations. Our study results suggest that, in certain subgroups of at-risk populations (e.g., tobacco users), and in the presence of neoantigens, the probability of detecting *TP53* variants in circulation will increase. This serves as a practical application of this study. With due caution, in light of the small study numbers, *TP53* mutations in serum cfDNA from tobacco users and mutations with a higher frequency of neoantigen formation were more likely to be detected.

## 4. Materials and Methods

### 4.1. Ethical Approval

This study was approved by the ethical committee of the Digestive Disease Research Institute of Tehran University of Medical Sciences, Tehran, Iran (IRB00001641, 11 January 2003), and Institutional Review Board of National Cancer Institute, Bethesda, MD, USA (NCT00339742, 25 March 2003), and IARC Ethics Committee (project No. 17–30).

### 4.2. Study Population, Sample Selection

The main study’s details were reported earlier [9,35]. Briefly, case subjects were recruited at Atrak clinic, the only specialized clinic for esophageal cancer diagnosis in eastern Golestan, from 2003 to 2008. Included cases were histopathologically confirmed ESCC patients who underwent upper gastrointestinal endoscopy and agreed to participate in the study. Biopsy specimens were oriented, fixed in 10% buffered formalin, embedded in paraffin, sectioned, stained with hematoxylin and eosin, and examined by experienced pathologists (M. Sotoudeh and B. Abedi-Ardekani). Two population-based control subjects were selected, individually matched to the cases by age (±2 years), sex, and location (i.e., from the same neighborhood or village). In total, we recruited 300 cases and 571 matched controls. After obtaining written informed consent, a nurse and a physician administered a structured questionnaire. No proxies were used. Data were collected on demographic variables, lifelong history of tobacco, opium, alcohol use, medication, and several potential confounders. Before endoscopy, a 12 mL venous blood sample was collected from each case subject. The serum was immediately separated into 5 mL EDTA-contained tubes, aliquoted, and stored at −80 °C. Collected blood samples from matched controls were transferred on ice in a cooler box (~4 °C). The time between collection and processing of neighborhood control samples was <12 h. This time interval was significantly higher for controls than ESCC cases, which was unavoidable due to the lack of laboratory facilities in the more than 200 villages visited while collecting neighborhood controls. We followed-up on cases and controls for the current study through a linkage to local cancer registry data updated until 2020. We also tried contacting neighborhood controls by phone to investigate their medical status.

Details of the analysis of *TP53* mutations in ESCC patients have been reported elsewhere by Abedi-Ardekani et al. [9]. In brief, DNA from formalin-fixed paraffin-embedded biopsy tissues with tumor purity of >70% was extracted using QIAamp DNA MicroKit from Qiagen (Hilden, Germany). For a total of 119 ESCC cases, the *TP53* coding region (exons 2–11) went through sequencing using Applied Biosystems PRISM 3100 Genetic Analyzer (Applied Biosystems, Foster City, CA, USA) in 2011. In total, 120 *TP53* mutations were detected in 107 ESCC cases. Of which 101 mutations were in exons 5–8.

Out of 107 ESCC cases with available data on tumor *TP53* mutations, we selected 30 ESCC cases based on an in silico experiment. We used a database of *TP53* variants in cfDNA of a series of patients with small-cell lung cancer to determine the error rate and proportion of false-positive detection per each genomic concordance [28]. We have added 12 further cases with more than one *TP53* mutations in tumors to our selected cases. We included 39 matched controls with available serum samples in our assay.

### 4.3. CfDNA Sequencing

CfDNA was extracted from less than 1 mL serum using QIAamp DNA circulating Nucleic acid kit (Qiagen, Hilden, Germany). Twenty-seven amplicons with 81 to 139 bp sizes were designed (Eurofine Genomics, Ebersberg, Germany) to cover the coding region of *TP53* (exons 2–11) and splicing sites. We prepared two plates in duplicate, containing equal molar DNA of 5 ng. We vacufuged 24 samples and concentrated DNA to reach the 10 uL reaction volume. Then, plates were dried out through heating at 65 °C for 2 h. We used GeneRead DNAseq Panel PCR kit V2 (Qiagen, Paris, France) for multiplex PCR to enrich the targets. PCR Mix consisted of 2 uL buffer, 1 uL primer pool (0.6 µM), and 6.27 uL water. PCR reaction was 95 °C (15 min) and 29 cycles of 95 °C (15 s), 60 °C (2 min), and 72 °C (10 min). Primers were diluted to 100 µM each and pooled in an equimolar way (1.85 µM final concentrations).

We used two quality controls: (1) a positive plasma control with known *TP53* mutation in cfDNA (Lung cancer sample RS113032 with Chr17:7576897G>A variant); (2) a pooled serum of 15 ESCC cases were divided into three duplicates (6 samples) without prior knowledge about the mutational profile of cases.

After amplification, we purified 10 uL of PCR product with 18 uL Serapure beads (ThermoFisher Scientific, Strasbourg, France). After purification, eight random samples were quantified using Qubit HS-ds DNA kit (Invitrogen, ThermoFisher Scientific, Strasbourg, France). The mean of DNA was 10 ng/uL. NEBNext Library Prep Set for Ion Torrent (New England BioLabs, Paris, France) with an in-house made P1 adaptor was prepared following manufacturer’s instructions. At this stage, individual barcodes were added, and amplicons were end-repaired and ligated to the adaptors. An additional step of bead purification with 1.8 ratios was carried out. A brief step of 4 cycles of amplicon amplification was performed through PCR reaction of 93 °C (30 s), 98 °C (10 s), 58 °C (30 s), and four cycles of 65 °C (30 s) and a final step of 5 min at 65 °C. Total PCR reaction volume was 25 uL, including 11 uL DNA. The second step of DNA quantification was done on another set of 8 random cases, and all samples had ranges between 11–17 ng/uL DNA. A total of 22 uL volume per well was left after purification and end-repair. All samples were pooled in an equimolar (assuming 10 ng/uL DNA in each well). We mixed 60 uL of pooled libraries with 12 uL of 6× loading dye and loaded those in 5 lanes of 2% agarose gel (20 uL per lane) for size selection. We ran the gel at 150 v for 90 min. After migration, bands with a size of 200 bp to 300 bp were cut and purified using MiniElute gel extraction kit (Qiagen, Paris, France). To verify the process of size selection and dimers’ absence, we ran 10 samples on the Bioanalyzer 2100 platform (Agilent Technologies, Santa Clara, CA, USA). Libraries were enriched using emulsion PCR on Ion Sphere particles followed by magnetic bead purification. Libraries were deep sequenced, targeting 10,000 depth on the Ion Torrent^TM^ Proton Sequencer using Ion TM Hi-Q^TM^ Sequencing 200 kit and Ion PI v3 (Thermo Fisher Scientific, Waltham, MA, USA).

### 4.4. Sequencing Data Analysis

We used Needlestack, a multi-sample variant caller designed for the detection of low abundance mutations. Details of the algorithm were mentioned in its methodological paper [28]. Briefly, needlestack estimated the sequencing error rate, at a particular position for a particular base change, using a negative binomial regression (NB) [36]. Then, it detected the true DNA mutations as being outliers of this model of errors. In this method, for each variant, a *p*-value would be dedicated to measuring the probability that a variant is part of the sequencing error model (regression outlier) (Appendix A). Due to the requirement of multiple samples to estimate this error model, the individual *p*-value was finally corrected for multiple testing and presented as Q-value in the Phred scale. In order to boost the precision of our method, and because Needlestack should correct only for sequencing artefacts, we used post-calling filters to remove potential remaining errors. These filters included: strand bias variants using relative variant strand bias (RVSB) < 0.85, and removing adjacent 10× lower allelic fraction variants within 5 nt region upstream or downstream of the candidate variants in order to remove artefacts from the alignment step. Figure 5 shows the proportion of losing variants after applying these filtering steps.

We based our variant detection in cfDNA on the agreement of two or more data points on the presence of mutations to increase the confidence of detecting true positives. Our data points consisted of prior knowledge of mutations in tumors and/or variants’ presence in both technical duplicates of the same sample. Two analysis methods were applied: (1) using a priori knowledge of *TP53* mutations in tumor FFPE samples and searching for the same mutations in cfDNA variants of cases and controls; (2) ignoring the known mutations in ESCC tumors in our series of cases and searching for duplicate mutations in cases and controls. We used a list of *TP53* driver mutations from a large case-series of ESCC tumors in endemic areas (480 ESCC cases) and examined the matched cfDNA variants in cases and controls for the sensitivity analysis. The fraction of observed identical genomic coordinates (position and base change) in the same individual tumor and cfDNA to all tumor mutations was considered a concordance fraction of cfDNA variant detection. Nonparametric statistical tests were done using Stata 14 (StataCrop. LLC, College Station, TX, USA). Cox regression model and log-rank tests were applied for survival analysis.

## Figures and Tables

**Figure 1 ijms-22-05627-f001:**
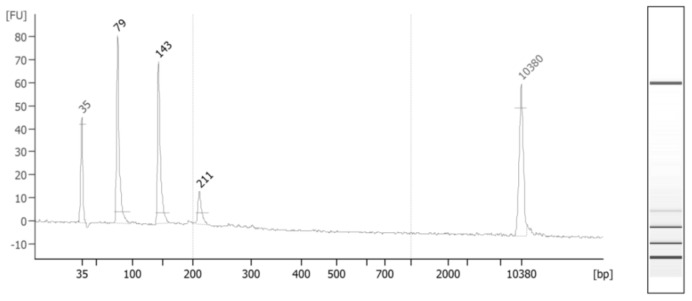
CfDNA fragments extracted from the serum of a randomly-selected ESCC case. The average size is 220 bp, showing several fragments under 200 bp (start and end picks are the standard markers). Long fragments (between 200 and 1000 bp) consisted of 11% of total serum cfDNA.

**Figure 2 ijms-22-05627-f002:**
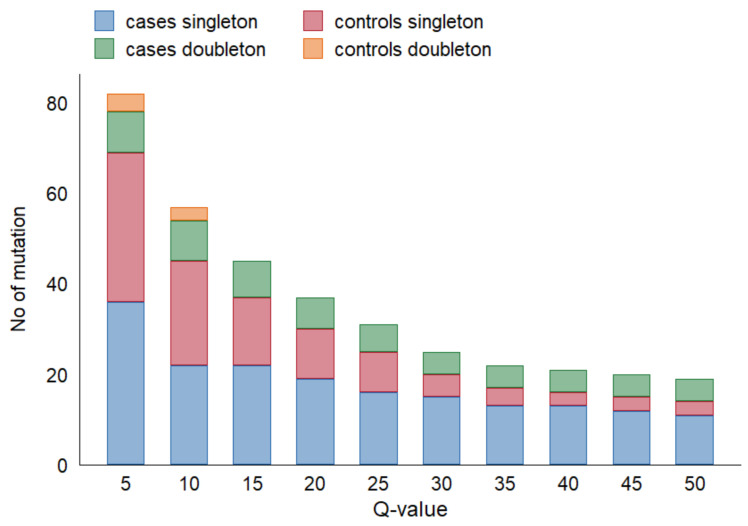
Effect of Q-value threshold on detecting *TP53* variants in cfDNA.

**Figure 3 ijms-22-05627-f003:**
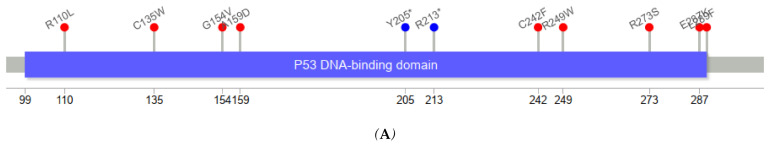
Visualization of mutation (mutations that have no record for protein data not included) [16] (**A**) *TP53* variants detected in cfDNA of ESCC cases concordant with a mutation in tumor tissue (R110L, C135W, G154V, A159D, Y205*, R213*, C242F, R249W, R273S, E287K, L289F) (**B**) *TP53* variants in tumor tissues not detected in cfDNAs of ESCC cases (P77T, P77P, T155P, I162I, V173M, V173L, H178P, R196*, V197G, S215R, V216M, Y220N, E224E, N235N, G244C, R249M, G266E, V272L, V272G, E286K, E287*, R306*) Appendix A contain the observed mutations in cfDNA and tumor tissues. Nonsynonymous mutations (red circle), synonymous mutation (blue circle).

**Figure 4 ijms-22-05627-f004:**
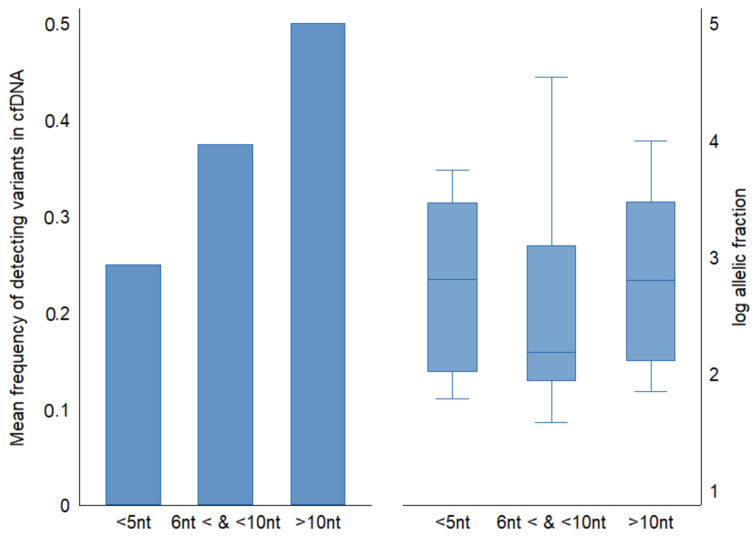
Comparison of frequency of detecting concordant variants in cfDNA relative to nucleotide distance between genetic coordinates of mutations in FFPE tumor tissue. Half of ESCC tumor *TP53* mutations consisted of adjacent variants (<10 nt apart from the next mutation). Allelic fractions across groups were similar.

**Figure 5 ijms-22-05627-f005:**
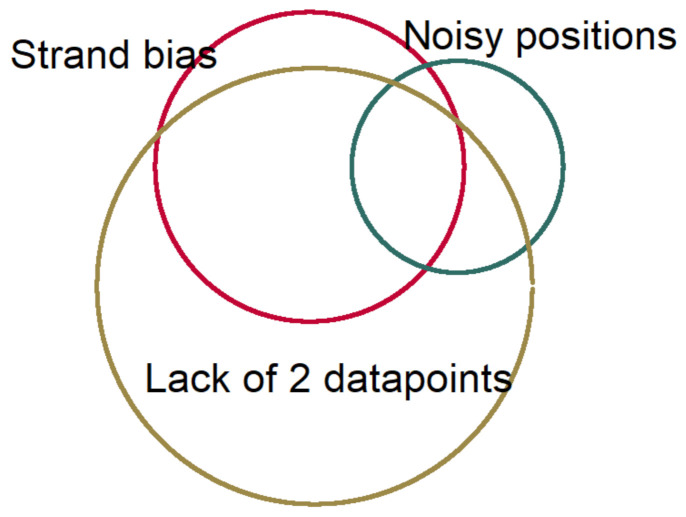
The relative proportion of losing data in each filtering step.

**Table 1 ijms-22-05627-t001:** Characteristics of cases and controls with and without concordant *TP53* cfDNA variants, comparing *TP53* mutations in FFPE tumor tissues.

Variables	ESCC Cases			Healthy Controls		
	cfDNA+	cfDNA−	*p*-Value	cfDNA+	cfDNA−	*p*-Value
Number	14 ^1^	29		5 ^1^	33	
Mean age(SD)	62 (9)	63 (11)	0.72	69 (16)	63 (9)	0.23
Sex (Female %)	44%	45%	0.75	57%	59%	0.25
Ever-smoker (%)	45%	16%	0.02	22%	18%	0.52
Chewing tobacco (%)	27%	10%	0.07	0%	3%	0.06
Ever-opium user (%)	35%	26%	0.49	28%	15%	0.41
Median cfDNA (ug)	179	72	0.02	165	139	0.72
Mean sequencing coverage	670,174	550,847	0.02	668,927	533,174	0.10

^1^ Healthy control who developed ESCC after the enrollment regrouped as a new case.

**Table 2 ijms-22-05627-t002:** Odds of detecting *TP53* mutations in cfDNA relevant to the frequency of neoantigen formations resulted from mutations.

Neoantigen Formation Frequency	Protein Variants from Detected Mutations in cfDNA	Protein Variants from Undetected Mutations in cfDNA	Unadjusted OR (95%CI)	Adjusted OR for HLA Frequency (95%CI)
1st quantile	71 (49%)	74 (51%)	Reference	Reference
2nd quantile	130 (55%)	106 (45%)	1.28 (0.84–1.93)	1.29 (0.85–1.95)
3rd quantile	123 (53%)	109 (47%)	1.17 (0.77–1.78)	1.22 (0.80–1.85)
4th quantile	131 (61%)	84 (39.1%)	1.62 (1.06–2.48)	1.91 (1.11–3.29)

**Table 3 ijms-22-05627-t003:** Characteristics of tumor TP53 variants according to detectability of variants in cfDNA cases and controls.

Variant Characteristics	ESCC Tumor *TP53* Variants		
	Detected in Case cfDNA	Undetected in Case cfDNA	Detected in Control cfDNA
Unique variants	14	29	5
Duplicate variants	1	5	0
Variant categories			
intronic	1 (7%)	0	0
missense	10 (72%)	25 (72%)	5 (100%)
nonsense	3 (21%)	4 (11%)	0
silent	0	3 (8.5%)	0
splice	0	3 (8.5%)	0
Variants in hotspots	12 (86%)	21 (70%)	4 (80%)
Mean REVEL score (SD)	0.87 (0.1)	0.84 (0.2)	0.87 (0.2)
Variants at cpg cites	4 (29%)	5 (14%)	1 (20%)
Variants at nucleosome main peak positions [18]	10 (45%)	19 (58%)	0
Mutation type			
A:T>C:G	2 (14%)	6 (17%)	1 (20%)
A:T>G:C	0	1 (3%)	0
A:T>T:A	1 (7%)	1 (3%)	0
G:C>A:T	6 (43%)	18 (51%)	3 (60%)
G:C>C:G	1 (7%)	0	0
G:C>T:A	4 (29%)	9 (26%)	1 (20%)

## Data Availability

Not applicable.

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
