# Peer review of "TP53 Targeted Deep Sequencing of Cell-Free DNA in Esophageal Squamous Cell Carcinoma Using Low-Quality Serum: Concordance with Tumor Mutation"

_ijms, 2021, doi:10.3390/ijms22115627_

Round 1

Reviewer 1 Report

The authors analyzed the possibility of using archival serum for detecting circulating cell-free tumor DNA. This is an important question because archival serum is often the only available form of liquid biopsy in most biobank and epidemiology studies and there is a lack of such data for answering the question.

The author applied a commercial targeted sequencing assay with sequencing replicates and careful data filtering. The conclusions are supported by the results. The data provides a valuable reference for future epidemiological studies aiming to analyze archival serum samples.

Author Response

April 27, 2021

Dear Editor-in-chief,

We would like to thank you and your colleagues for reviewing our manuscript entitled:” TP53 targeted deep sequencing of cell-free DNA in esophageal squamous cell carcinoma using low-quality serum: concordance with tumor mutation”. We have revised the manuscript as instructed by the reviewers, and have given a point-by-point response to reviewers’ comments (please see below). We trust these changes have improved our manuscript and we hope that the manuscript is now acceptable for publication.

Sincerely,

Behnoush Abedi-Ardekani, MD, Pathologist (anatomical pathology & laboratory medicine),

Office : +33472738 504, E-mail: [email protected]

Dariush Nasrollahzadeh, MD, PhD (postdoctoral fellow), E-mail:[email protected]

Genomic Epidemiology Branch, International Agency for Research on Cancer,

World Health Organization, 150 Cours Albert Thomas, 69372 Lyon, France

************************************************************************

Referee #1

************************************************************************

Referee 1-1) The author applied a commercial targeted sequencing assay with sequencing replicates and careful data filtering. The conclusions are supported by the results. The data provides a valuable reference for future epidemiological studies aiming to analyze archival serum samples.

 Reply: We appreciate reviewer’s positive feedback and kind comment.

Reviewer 2 Report

The manuscript entitled “ TP53 targeted deep sequencing of cell-free DNA in esophageal squamous cell carcinoma using low-quality serum: concordance with tumor mutation” investigates the concordance of TP53 mutations in tumor tissue and cfDNA extracted from archival serum left-over among 42 cases and 39 matched controls (age, gender, residence) in a high-risk area of Northern Iran (Golestan). Deep sequencing of TP53 coding regions was complemented with a specialized variant caller for rare variant detection (Needlestack). Overall, 23% to 31% of mutations were concordantly detected in tumor and serum cfDNA (based on two false discovery rate thresholds). Concordance was positively correlated with high cfDNA concentration, smoking history (p-value=0.02) and mutations with a high potential of neoantigen formation (OR-95%CI=1.9 (1.11-3.29)). TP53 mutations were identified in five controls, one of whom was subsequently diagnosed with ESCC.

The followings need to be addressed.

1.Studies have found that compared with healthy people, cfDNA has a higher concentration in patients with inflammatory diseases such as metastatic cancer, trauma, myocardial infarction, autoimmune diseases or sepsis et al. Why is the cfDNA concentration in ESCC cases (1.67 ng/ml) lower than controls (3.12 ng/ml)?

2. In the sentence “A total of 63% of ESCC cases with detected concordant ctDNA and 29% of 135 cases without detectable concordant ctDNA were either tobacco or opium users (p-136 value=0.04)”, what dose ctDNA mean? Please list the whole name.

3. In the Table 1, it has a high percentage of cfDNA- in ESCC cases. Moreover, as authors describe “70% non-identifiable mutations among cancer patients and 12% mutation 38 detection in controls are strong limitations for using serum cfDNA for screening the entire coding sequence of a gene”. It seems that the method focused in this study has a large limitation in clinical application. What kind of contribution or impact thought by authors can this study provide?

Author Response

************************************************************************

Referee #2

************************************************************************

Referee 2-1)The followings need to be addressed.

Reply: We appreciate reviewer for his/her comments, here please see our replies:

Referee 2-2) Studies have found that compared with healthy people, cfDNA has a higher concentration in patients with inflammatory diseases such as metastatic cancer, trauma, myocardial infarction, autoimmune diseases or sepsis et al. Why is the cfDNA concentration in ESCC cases (1.67 ng/ml) lower than controls (3.12 ng/ml)?

Reply: We thank reviewer for his/her comment. We have modified the text and clarified the presence of outliers among controls in results, and explained the timing between withdrawing blood and performing centrifuge between clinical cases and neighbourhood controls as the possible source of higher level of lysis among controls in methods and discussion sections. We emphasized lack of statistical difference in total cfDNA between cases and controls after removing outliers from controls in results:

 We have added following to the results section:

“(Result section-page 3 lines 106-109) …after excluding 3 outliers for cfDNA level (> 100 ng/ml) the mean concentration of cfDNA among neighbourhood controls and cases were 1.62 and 1.18 ng/ml (p-value=0.06).

We have expanded discussion part:

“(discussion section-page 8 lines 229 -235) … Due to the matched study design, pre-analytical conditions for cases and controls were almost similar. The only significant difference was the longer time interval between withdrawing blood and performing centrifuge for neighbourhood controls compared to cases in research clinic. The longer time-interval was due to shipment of blood in cooler from distant villages to research centre. On road shipment and longer time prior to serum extraction, might cause more lysis and as a result we observed marginally but statistically non-significant higher level of cfDNA among neighbourhood controls compared to ESCC cases.

We included below text in Methods section:

“(Method section-page 10 line 380)… This time interval was significantly higher for controls than ESCC cases, although unavoidable due to the lack of lab facility in more than 200 visited villages for collecting neighbourhood controls”

 Referee 2-3): In the sentence “A total of 63% of ESCC cases with detected concordant ctDNA and 29% of 135 cases without detectable concordant ctDNA were either tobacco or opium users (p- value=0.04)”, what dose ctDNA mean? Please list the whole name.

Reply: We thank the reviewer for the comment. We omitted “ctDNA” and adapted cfDNA in the text instead. (Page 4, lines 145-146)

Referee 2-3) In the Table 1, it has a high percentage of cfDNA- in ESCC cases. Moreover, as authors describe “70% non-identifiable mutations among cancer patients and 12% mutation 38 detection in controls are strong limitations for using serum cfDNA for screening the entire coding sequence of a gene”. It seems that the method focused in this study has a large limitation in clinical application. What kind of contribution or impact thought by authors can this study provide?

Reply: We appreciate reviewer’s comment. We have modified the text, highlighting that our study is the first attempt using whole coding region of the most commonly observed driver gene in ESCC (TP53) and result is comparable with single gene screening in other cancers or comparable with using several point mutations instead. We echoed knowledge gaps reported by other cfDNA studies and added newly reported limitations for early detection studies and emphasized on the practicality of our finding through underlining neoantigens, adjacent mutations and risk factor profile:

We have expanded conclusion and added:

“(Discussion, page 10: line 339-352)…. The implications of our findings are important echoing the message for mutation-based cancer biomarkers when whole coding regions are blindly screened. This limitation is beyond selection of the type of biologic samples (serum or plasma). At the same time we have shown the importance of ultra-sensitive rare variant caller to avoid recruiting false positive results. We have also portrayed the biologic challenges such as adjacent variants, neoantigens, and certain carcinogenic profiles influencing the specificity of mutation detection in cfDNA. This study as the first attempt for screening whole coding regions of TP53amongst one of the highest incident areas for ESCC reached similar results to those applied same method in different organs and non-endemic populations. Our study suggests that in certain subgroup of at –risk population (tobacco users), and in presence of neoantigens, the probability of detecting TP53 variants in circulation will increase, which serves as a practical application of this study.

We have relocated this sentence to the abstract and modified abstract:
“(Abstract, page 1: line  35): …. suggesting that tumor DNA release in the bloodstream might reflect the effect of immune and inflammatory context on tumor cell turnover.

We modified the abstract’s conclusion:

“(Abstract, page 1: line 43-45): …Yet, 70% non-identifiable mutations among cancer patients and 12% mutation detection in controls are main challenges in applying cfDNA to detect cancer-related variants when blindly targeting whole coding region of driver TP53 gene in ESCC.

Round 2

Reviewer 2 Report

The authors have answered my questions and revised the manuscript accordingly.